# Conformational Stability of Poly (N-Isopropylacrylamide) Anchored on the Surface of Gold Nanoparticles

**DOI:** 10.3390/ma14020443

**Published:** 2021-01-18

**Authors:** Runmei Li, Cong Cheng, Zhuorui Wang, Xuefan Gu, Caixia Zhang, Chen Wang, Xinyue Liang, Daodao Hu

**Affiliations:** 1Engineering Research Center of Historical and Cultural Heritage Protection, Ministry of Education, School of Materials Science and Engineering, Shaanxi Normal University, Xi’an 710062, China; lrm@snnu.edu.cn (R.L.); congcheng2017@snnu.edu.cn (C.C.); wangzhuorui@snnu.edu.cn (Z.W.); zcx@snnu.edu.cn (C.Z.); ccw1128@snnu.edu.cn (C.W.); 2College of Chemistry and Chemical Engineering, Xi’an Shiyou University, Xi’an 710065, China; XuefanGu@xsyu.edu.cn; 3Institute of Industrial Hygiene of Ordnance Industry, Xi’an 710065, China; ancybz@163.com

**Keywords:** gold nanoparticles, PNIPAM, immobilized PNIPAM, failed thermo-sensitivity

## Abstract

To verify the temperature sensitive failure of poly (N-isopropylacrylamide) (PNIPAM) anchored on the surface of gold nanoparticles (AuNPs), the UV-Vis spectra with temperature variations of the following aqueous solutions respectively containing AuNPs-PNIPAM, Au-PNIPAM/PNIPAM, PNIPAM, in different media (including salt, ethanol, HCl and cetyltrimethylammoniumbromide (CTAB)), were systematically determined. The results indicated that the UV-Vis spectrum of AuNPs-PNIPAM suspension hardly changed even above the Lower Critical Solution Temperature (LCST) of PNIPAM, but that of Au-PNIPAM/PNIPAM sharply increased only in absorbance intensity. A possible mechanism of the failed temperature sensitivity of PNIPAM anchored on the surface of AuNPs was proposed. Being different from free PNIPAM molecules, a strong interaction exists among PNIPAM molecules anchored on the surface of AuNPs, restraining the change in conformation of PNIPAM. The temperature sensitivity of Au-PNIPAM/PNIPAM originates from the free PNIPAM molecules rather than the anchored PNIPAM one. The changing electrostatic interaction could effectively regulate the aggregation behavior of AuNPs-PNIPAM and enhance its sensitivity to temperature.

## 1. Introduction

Gold nanoparticles (AuNPs) are of particular interest to researchers due to their unique optical and electronic properties [1]. AuNP solutions exhibit a strong surface plasmon resonance (SPR) due to the collective oscillation of the conduction electrons across the nanoparticle and display intense size-dependent color in the visible spectrum [2]. For various applications, the surface coating of AuNPs is generally essential. Smart polymer coating allows AuNPs to sense external stimuli, including temperature, pH, and light irradiation. Among these smart polymers, poly (N-isopropylacrylamide) (PNIPNAM) is often used for coating AuNPs to produce a core-shell system whereby the AuNPs are encapsulated in a thermo-responsive polymer shell. Such composites have been shown to have interesting applications in drug delivery [3,4], photothermal therapy and imaging [5], chemical detection [6], catalytic oxidation [7], and other biologically relevant applications [8,9,10]. As we all know, PNIPAM undergoes a phase transition in pure water from hydrophilic, fully hydrated polymer chains to hydrophobic-collapsed chains at temperatures around 30–34 °C. Upon heating, PNIPAM becomes insoluble and precipitates from the solution, and the temperature at which this occurs is referred to as a lower critical solution temperature (LCST) [11]. Thus, PNIPAM chains immobilized on the surface of AuNPs (AuNPs-PNIPAM) are also naturally expected to show a similar thermo-responsiveness and thus result in the AuNPs-PNIPAM solution exhibiting a reversible color change. However, the actual findings in the published literature have shown that the change in color of AuNPs-PNIPAM is often different from that expected. Most probably, it is an oversimplified idea that the temperature-sensitive color change of AuNPs could be realized by means of AuNPs-PNIPAM. In fact, how the color of AuNPs-PNIPAM changes with temperature is related to not only the characteristics of immobilized PNIPAM but also the property of AuNPs.

Firstly, the electrostatic repulsion of the negatively charged AuNPs strongly prevent from undergoing the aggregation of AuNPs-PNIPAM and the results of zeta potential measured have confirmed this suggestion [12,13,14,15]. In these reports, while the view that electrostatic repulsion restrains AuNPs-PNIPAM aggregation is consistent, there are different views as to whether the conformational change of the immobilized PNIPAM occurs above LCST. According to the quasi-elastic light scattering results, S. Yusa et al. suggested that PNIPAM anchored on the surface of AuNPs could collapse into a compact form when the temperature was above LCST [12]. A similar conclusion was drawn in the literature [14,15]. R. Hoogenboom and co-workers pointed out the tethered PNIPAM did not change in conformation above LCST because both the values of particle size and position of the plasmon peak as functions of temperature are constant [2,13]. Furthermore, the electrostatic repulsion of AuNPs depends on the ionic strength [15,16]; thus, the effect of salt on the behavior of the aqueous dispersion of AuNPs-PNIPAM were widely investigated. However, in the literature, there are some different findings about this effect. S. Yusa et al. found that the color variation for AuNPs-PNIPAM in the presence of NaCl was reversible, and they attributed this reversibility to the temperature-dependent reversible association among AuNPs-PNIPAM particles [12]. A similar finding was also reported by R. Hoogenboom et al. [14] and Bruno G. De Geest et al. [13]. Differently, W. Cheng et al. [17] found that the effect of NaCl on the aggregation of AuNPs-PNIPAM in thermo-cycling is poorly reversible.

Secondly, in the literature, there are different findings as to whether the anchored PNIPAM molecules still have the same temperature sensitivity as the free ones. Alexander D. Q. Li and co-workers [18] reported that AuNPs-PNIAPM has a reversible thermo-sensitivity in a clear-opaque transition between 25 and 30 °C, and this transition is sharper than that of free PNIPAM, and they attributed this difference to the PNIPAM interchains self-association within a single nanoparticle driving the surface of the nanoparticle to switch from hydrophilic to hydrophobic. However, O. A. Scherman et al. reported that the aggregation of AuNPs-PNIPAM caused by temperature is likely due to the presence of free PNIPAM molecules rather than the anchored PNIPAM [19]. This finding evidently implies that the property of immobilized PNIPAM is different from that of the free one. This viewpoint is also supported by various reports. The UV-vis absorption spectrum of the AuNPs-PNIPAM solution upon an increase in temperature above LCST does not show any significant change of the SPR band [12,14,20] and the slight red shift may be attributed to a change in the dielectric constant of the microenvironment around the gold nanoparticles rather than thermo-sensitivity of PNIPAM.

Based on summarized relevant researches on thermo-sensitivity of AuNPs-PNIPAM, the following consensus conclusions could be drawn. Besides the electrostatic repulsion, the immobilization of PNIPAM is also an important factor in the thermo-sensitivity of AuNPs-PNIPAM. Clearly, the difference in temperature sensitivity between anchored and free PNIPAM is due to their different molecular flexibility. Namely, the effect of the interaction among the anchored PNIPAM molecules on the temperature sensitivity of PNIPAM could not be ignored. Petra Uhlmann found that the interaction among the immobilized PNIPAM molecules is so strong that even at temperatures above the LCST, PNIPAM palisades still contain intermolecular hydrogen bonds between segments and water, and PNIPAM molecules in higher grafting density still adopt more brush-like conformation above the LCST [21]. In fact, unusual thermo-temperature behavior of immobilized PNIPAM has received special attention [21,22,23,24,25]. The interchain interaction among the anchored PNIPAM molecules hinders their de-swelling, and the swelling ratio of immobilized PNIPAM is found to decrease noticeably at high densities [25]. Plunkett et al. also pointed out that the ability to thermally tune anchored PNIPAM molecules depends strongly on the molecular weight and grafting density of PNIPAM due to the non-negligible interaction between PNIPAM, and this interaction makes the anchored PNIPAM molecules adopt a two-layer structure with a compressible outer phase and a dense, incompressible inner phase [23]. A similar view was reported by Tenhu et al. that the PNIPAM chains in the inner zone undergo the first transition at a lower temperature, and that in the outer zone show the second transition at a higher temperature [26]. Additionally, this interaction is also affected by the size of AuNPs because the average volume per chain for surface grafted PNIPAM molecules is dependent on the radius of AuNPs [27]. In fact, a suitable immobilized density of PNIPAM is very important for AuNPs-PNIPAM to have a visibly macroscopic change in thermo-sensitivity. Too low immobilization density is not enough to produce visible macroscopic changes, and too high immobilization density could not provide enough room for conformation changes [28].

Based on the above-mentioned reports in the literature, it is concluded that the conformational stability of surface-immobilized PNIPAM molecules on AuNPs remains a controversial issue. The complicity in the nanocomposite system of AuNPs-PNIPAM is the main reason for these inconsistencies that existed in the reported literature. There are many factors impacting the UV-Vis spectrum of AuNPs-PNIPAM, such as the length of polymers, the modification density, the size and morphology of AuNPs. In principle, it is necessary to take these factors into full account in this study. Objectively speaking, however, it is hard to consider all the above factors in an article. Therefore, in this work, to verify the conformational stability of PNIPAM in the color change of AuNPs-PNIPAM, we just take a special system, given 20 nm of AuNPs saturated loaded with PNIPAM, as a case, to be expected to obtain inspiration. Since the SPR of AuNPs is sensitive to the local environment, Uv-vis spectroscopy is an effective means to detect the surroundings of AuNPs. We used the UV-vis absorption spectroscopy to study the effects of temperature, salt, ions and solvent diffusion on the aggregation of AuNPs-PNIPAM. The findings indicated that the interaction among the immobilized PNIPAM molecules has a strong inhibiting effect on their conformational change, and the aggregation of AuNPs-PNIPAM particles caused by salt or HCl is mainly attributed to the electrostatic effect and local refractive index change rather than the conformational change of PNIPAM. Namely, the temperature sensitivity of PNIPAM immobilized on the AuNPs surface is difficult to function. This conclusion suggests researchers who expect to develop AuNPs-PNIPAM-based sensors through the conformational change of PNIPAM should take this inspiration into account. Although the system studied in this paper is only a typical one, it has a universal reference significance for the system of AuNPs intensively modified with PNIPAM. These findings are important not only for clarifying the inconsistent results reported in the literature but also for designing functionalized AuNPs-PNIPAM sensing systems.

## 2. Experiment Section

### 2.1. Materials

Tetrachloroauric (HAuCl_4_·4H_2_O, purity ≥ 99.9%), ethyl alcohol, and chlorhydric acid (HCl) were purchased from Sinopharm Chemical Reagent Co., Ltd (Shanghai, China). NaCl, Cetyltrimethylammoniumbromide (CTAB) and trisodium citrate (AR) were received from TIANLI Chemical Reagents Ltd. (Tianjin, China). N-isopropylacrylamide (NIPAM), 4, 4′-azobis (4-cyanopentanoic acid) (ACPA), 4-cyano-4-(thiobenzoylthio)-pentanoic acid (CPD), tris (2-carboxyethyl) phosphine hydrochloride (TCEP), 1,4-dioxane were obtained from Aldrich (St. Louis, MO, USA). All reactants were analytical grade and used without further purification except NIPAM and 1,4-dioxane. NIPAM was recrystallized in a mixture of benzene and n-hexane (3:7, v/v) prior to use. 1,4-dioxane was treated by Na_2_SO_3_ aqueous solution and then by high concentration of HCl, and then was dried by KOH and distilled in the presence of metal Na before using. Milli-Q water was used in all the experiments. All used glassware was treated with aqua regia, rinsed in Milli-Q water, and oven-dried prior to use.

### 2.2. Preparation of AuNPs Coated with PNIPAM-SH

#### 2.2.1. Preparation of AuNPs

Gold nanoparticles stabilized by citrate were synthesized via a previously reported method [29]. Briefly, 1 mL HAuCl_4_ (25 mM) aqueous was injected into 150 mL of boiling solution sodium citrate (2.2 mM), and the solution was kept boiling for 10 min. This led to a series of color changes from purple, blue, bluish-gray and to red color indicating the formation of citrate-capped AuNPs used as the seed solution (~10 nm, ~3 × 10^12^ NPs/mL). Then the solution was cooled down to 90 °C and 2 mL of the obtained solution was subtracted. The final step was the addition of 1 mL of HAuCl_4_ solution (25 mM) to the left seed solution and keeping the temperature of the solution constant at a temperature of 90 °C for 30 min. After the final step was repeated once again, the reaction was quenched by an ice-water bath. This process produces AuNPs with a size of ~20 nm. The TEM image and size distribution are shown in Figure 1. The concentration of the as-prepared AuNPs was approximately the same as the seed particles (~3 × 10^12^ NPs/mL).

#### 2.2.2. Preparation of PNIPAM-SH

PNIPAM-SH was prepared following a reported procedure [30] and the detailed synthetic route can be seen in Appendix A. CPD (0.0444 g, 158.5 μmol) and ACPA (0.018 g, 63.8 μmol) were added to a NIPAM (0.2646 g, 20 mmol) solution of 1,4-dioxane (20 mL). The solution was degassed by purging with nitrogen gas for 30 min, and then heated at 70 °C for 24 h. After being cooled down in an ice bath, the reaction mixture was dialyzed against DI water at 4 °C for 3 days. After dialysis, the solution was freeze-dried to obtain the PNIPAM. To remove the terminal dithiobenzoate group, an aqueous solution of PNIPAM was treated with 2-ethanolamine (30 mol equiv of PNIPAM) and a trace amount (3–5 mg) of TCEP at 25 °C for 24 h. The solution was dialyzed against DI water at 4 °C for 3 days, and the PNIPAM-SH was then obtained after lyophilization. The composition characterized results (photo of sample, IR, NMR, GPC) of PNIPAM-SH could be seen in Appendix A.

#### 2.2.3. Preparation of Solutions Related to the Interaction with AuNPs

AuNPs solution used in all experiments was prepared by centrifuging repeated three times and washing of the synthesized AuNPs solution at 4 °C (19,000 rpm, 20 min), and denoted as AuNPs. 

AuNPs modified with PNIPAM-SH were prepared just by mixing and incubation of 50 mL AuNPs and 20 mg PNIPAM-SH powder for 48 h [19]. The resultant solution was denoted as the Au-PNIPAM/PNIPAM solution (in the presence of free PNIPAM). By centrifuging three times and washing the mixture at 4 °C (19,000 rpm, 20 min), the residue was diluted to the same volume and named by the AuNPs-PNIPAM solution (in absence of free PNIPAM) (the TEM image, size distribution and TEM Energy-dispersive X-ray spectroscopy (EDS) images and spectrum were shown in Figure 1), and the centrifugal supernatant in the first time as the free PNIPAM solution was used in the related studies.

In the examination of the salt effect, the concentration of NaCl was maintained at 16.9 mM without following instructions. To study the aggregation behavior of AuNPs-PNIPAM in different conditions, the following solutions were prepared. A mixture of 100 μL AuNPs-PNIPAM and 2 mL ethanol; a mixture of 2 mL AuNPs-PNIPAM and 2 mL CTAB (0.05 mM); a mixture of 1.2 mL AuNPs-PNIPAM and 300 μL HCl (0.01 M); PNIPAM solution (3.47 × 10^−6^ mg/L) and HCl solution (1 M) was prepared for the titration experiment.

### 2.3. Characterization

The TEM images for AuNPs and AuNPs-PNIPAM were obtained from a JEM-2100 instrument (JEOL, Tokyo, Japan) with an accelerating voltage of 200 kV (20 μL of the sample was dropped onto a carbon coated Cu grid and dried in air). TEM Energy-dispersive X-ray spectroscopy (EDS) spectra of Au-PNIPAM composites were collected with a JEM-2800 in STEM mode. The ^1^H NMR spectrum of PNIPAM-SH was monitored by a JNE-ECZ400S/L1 Nuclear Magnetic Resonance Spectrometer (400 MHz NMR, JEOL, Tokyo, Japan) using chloroform-d (CDCl_3_) as the solvent. Fourier transform infrared (FT-IR) spectroscopy was recorded in the range from 500 to 4000 cm^−1^ using a Perkin Elmer FT-IR spectrophotometer with attenuated total reflectance (ATR) mode. The average molecular weight of prepared PNIPAM-SH and molecular weight distribution (Mw/Mn) were determined via Waters 1515–2414 gel permeation chromatography at 25 °C in THF with a flow rate of 1 mL/min. UV-vis spectroscopy was performed using a Lambda 35 UV-Vis Spectrometer (PerkinElmer, Waltham, MA, USA). Zeta potential was monitored by a Malvern Zetasizer Instrument (Worcestershire, UK), and each sample was measured three times to obtain an average value.

## 3. Results and Discussion

Figure 1 presents TEM images for AuNPs (A) and AuNPs-PNIPAM (B). The prepared citrate-capped AuNPs are about 20 nm, and PNIPAM modified AuNPs are slightly larger than AuNPs. Their corresponding size distributions are shown in Figure 1C,D revealing their good monodispersity. Because of the strong penetrating capacity of the electron beam, the PNIPAM shell on the surface of AuNPs is hard to display in the TEM images. To testify for the PNIAPM shell loaded on the surface of AuNPs, Figure 1F–H gives TEM EDS mapping images and EDS spectrum for the gold element and nitrogen element, respectively. There is a matching spatial distribution for the nitrogen element and gold element, indicating the existence of PNIPAM and the core-shell structure of AuNPs-PMIPAM.

To reveal the effect of the interaction among the immobilized PNIPAM molecules of AuNPs- PNIPAM on their conformation changes, the effect of temperature, salt, ethanol, acid and CTAB on the SPR for AuNPs and AuNPs-PNIPAM were compared.

### 3.1. The Effect of Temperature on the Colorimetric Properties of AuNPs-PNIPAM

To examine the difference between AuNPs-PNIPAM and AuNPs in thermo-sensitive colorimetric properties, the changes of their UV-Vis spectra with temperature were monitored, and the resultant spectra are shown in Figure 2. For AuNPs-PNIPAM (Figure 2A), the spectra have a very slight change from room temperature to 90 °C, implying that there is no aggregation between AuNPs-PNIPAM in this case. This similar observation was also reported in the literature [12,13,14,17,19,20]. S. Yusa et al. suggested when the temperature was above LCST, AuNPs-PNIPAM did not associate with each other due to electrostatic repulsion of negative charges of the AuNPs [12]. This suggestion was confirmed by the measurement of the Zeta potential [13,14,15]. Besides the electrostatic repulsion of citrate-capped AuNPs, the interaction between immobilized PNIPAM also plays a positive role in the stability of AuNPs-PNIPAM. If the immobilized PNIPAM molecules were a hydrophobic compressed conformation at above LCST, it would inevitably lead to the AuNPs-PNIPAM of obvious aggregation. In this case, once the immobilized PNIPAM molecules formed a compression layer, it would lead to not only the obvious decrease in the electric double layer thickness of the AuNPs-PNIPAM but also the significant increase in the hydrophobicity of the AuNPs-PNIPAM. Both effects are conducive to the occurrence of aggregation. Therefore, we believe that the immobilized PNIPAM molecules have no conformational change at above LCST due to no significant aggregation of AuNPs-PNIPAM occurred above LCST. The slight change in the spectrum of AuNPs-PNIPAM with increasing temperature is only related to the escape of partial water molecules in the palisades of the immobilized PNIPAM molecules. As the literature reported, the slight red-shift in wavelength and the decrease in absorbance of SPR may attribute to the increase in the dielectric constant of the microenvironment around AuNPs [12,31]. Here, the increase in the dielectric constant is probably related to the escape of partial water in PNIPAM palisades. It is worth noting that the temperature of induced spectral change at above 70 °C in Figure 2 is much higher than that of the conventional LCST (32 °C) of PNIPAM, indicating that it is difficult for the change of the conformation associated with the water escape of the immobilized PNIPAM. As Petra Uhlmann and co-workers reported, PNIPAM palisades still contain intermolecular hydrogen bonds between segments and water, and PNIPAM molecules adopt more brush-like conformation above the LCST [21].

Moreover, Zeta potential data detected show that it seems unreasonable to interpret the thermal insensitivity of AuNPs-PNIPAM only in terms of electrostatic repulsion. In our experiments, the absolute value of AuNPs Zeta potential (about −48 mV) is truly higher compared with AuNPs-PNIPAM (−28 mV). In terms of electrostatic repulsion, the solution of AuNPs should be more stable than that of AuNPs-PNIPAM. The variation of the UV-Vis spectrum of AuNPs with temperature shown in Figure 2B, however, is more evident than that of AuNPs-PNIPAM. For the citrate-capped AuNPs, the SPR spectrum decreases in intensity and broadens, slightly red-shifts, when increasing the temperature from 25 to 90 °C. This special protective effect of PNIPAM to AuNPs is discussed in the following.

For citrate-capped AuNPs (Figure 2B), the sample used was obtained through centrifugal separation of the AuNPs solution prepared by the Turkevich synthesis method [32]. As a result, the citrate concentration in the separated solution is significantly reduced, the adsorbed citrate groups onto AuNPs are inevitably decreased. Especially, the desorption of citrate groups adsorbed on the surface of AuNPs is enhanced by heating. Consequently, the aggregation of AuNPs makes a broadening of the plasmon band accompanied by a decrease in intensity and a slight red-shift [1,33]. For AuNPs-PNIPAM, although the charge density of AuNPs-PNIPAM decreases due to charged citrate groups replaced by PNIPAM without charges, the densely immobilized SH-PNIPAM molecules onto the surface of AuNPs effectively resist AuNPs-PNIPAM aggregation induced by higher temperature. Thus, the above-mentioned special protective effect of PNIPAM to AuNPs should be related to the steric hindrance of the PNIPAM layer [14], hydrated PNIPAM palisades [21] and strong interaction among PNIPAM molecules [21].

Additionally, the broad peak located at 200–300 nm in the UV-Vis spectra changed with increasing temperature for AuNPs, while it barely changed for AuNPs-PNIPAM. The above spectral change may be related to the change of citrate. A detailed discussion of the UV spectra of AuNPs and AuNPs-PNIPAM is seen in Appendix A. This contrasting result further indicates the tethered PNIPAM improves the thermo-stability of AuNPs.

All the above-mentioned results indicated that PNIPAM molecules densely immobilized onto the surface of AuNPs possess the conformation stability even when the temperature far exceeds the LCST (32 °C) of free PNIPAM.

We also note that some research reported that AuNPs-PNIPAM solution in the presence of free PNIPAM has thermo-sensitivity. We believe that this thermo-sensitivity does not come from the immobilized PNIPAM but the free PNIPAM. To confirm this, the change of the spectrum with temperature for AuNPs-PNIPAM/PNIPAM was selected. The UV-vis spectra are shown in Figure 3A. Below 40 °C, the SPR characteristic peak at 520 nm is significant for AuNPs-PNIPAM/PNIPAM solution, and the spectra are almost constant. In the range from 40 to 70 °C, the SPR spectral intensity abruptly increases, while slowly increases above 70 °C. Additionally, the spectrum finally returns to the original location when the temperature falls to room temperature, presenting a reversible thermo-responsiveness. Throughout the process of this change, this reversibility does not be manifested in the shift of SPR maximum absorption wavelength but the absorption intensity. It implies that the reversible thermo-responsiveness originates from the conformation variation of free PNIPAM rather than the reversible aggregation change of AuNPs-PNIPAM. The UV-Vis spectrum of the supernatant after centrifuging AuNPs-PNIPAM/PNIPAM verifies this deduction. An expected result shows in Figure 3B. The spectral profile and variation trend with the temperature of the supernatant only containing PNIPAM are very similar to that of Au-PNIPAM/PNIPAM. The abrupt variation of the transmission in the inset (Figure 3B) is the same as that of PNIPAM near the LCST (32 °C). Obviously, thermo-responsiveness of Au-PNIPAM/PNIPAM is caused only by the free PINPAM rather than the immobilized PNIPAM. If the immobilized PNIPAM had a similar conformation transition characteristic to the free PNIPAM, the free PNIPAM would act like a cross-linker to facilitate the agglomerate of Au-PNIPAM. The change of spectra with temperature does not show the agglomerate for Au-PNIPAM/PNIPAM, implying that the conformation of anchored PNIPAM molecules fails to change with temperature.

All the above discussions are briefly summarized as follows. The conformational change of PNIPAM must be accompanied by a transition from hydrophilic to hydrophobic, which could result in the change of AuNPs-PNIPAM dispersion state and the increase of absorbance of Uv-vis spectrum over 600 nm. For the Au-PNIPAM system, which was absent of free PNIPAM, SPR did not present a distinct change, and in particular there was no obvious absorption peak above 600 nm (Figure 2A), indicating a failed conformational change. For the AuNPs-PNIPAM system in the presence of free PNIPAM, the Uv-vis spectrum exhibited a strengthened absorbance intensity and no distinct change in the SPR peak position (Figure 3A), and actually, the spectrum is just a superposition of the spectrum of non-temperature sensitive AuNPs-PNIPAM and that of temperature sensitive free PNIPAM. In addition, if the conformation of immobilized PNIPAM and free PNIPAM simultaneously changed when the temperature was higher than LCST, hydrophobic interaction between free and immobilized PNIPAM would inevitably lead to more significant aggregation of AuNPs-PNIPAM. However, no experimental results like the above assumption occurred, still suggesting a failed conformational change of immobilized PNIPAM.

### 3.2. The Effect of Salt on the Colorimetric Properties of AuNPs-PNIPAM

As previously mentioned, the addition of salt is an approach to reduce the electrostatic repulsion of AuNPs-PNIPAM, enhancing the aggregation of AuNPs-PNIPAM. However, in the literature, some reported that the agglomeration behavior of AuNPs-PNIPAM in the presence of salt is reversible [12,13,14] and some found that it is poor reversible [17]. It is well-known that an electrolyte added would make dispersed solutions of bare AuNPs unstable due to an electrostatic screen.

If the immobilized PNIPAM in the presence of salt had a conformation change when the temperature was above its LCST, the resulted hydrophobicity combining with a decrease in Zeta potential due to the displacement of citrate to PNIPAM would enhance the aggregation of AuNPs-PNIPAM. It would lead to a severe change of the SPR peak for AuNPs-PNIPAM compared with that of AuNPs. However, its SPR peak scarcely changes with temperature for AuNPs-PNIPAM in the presence of trace NaCl (Figure 4A) compared with that for the AuNPs counterpart (Figure 4B). Thus, we believe that whether reversible or irreversible aggregation of AuNPs-PNIPAM induced by the salt is mainly related to the charge decrease rather than the resulting conformation variation of tethered PNIPAM by temperature, which thus depends on the salt concentration [12]. As we expected, when the salt concentration increases (16.9 mM), the spectra changes with temperature whether for AuNPs-PNIPAM (Figure 4C) or AuNPs (Figure 4D) compared with their counterparts at lower salt concentration (3.2 mM) (Figure 4A,B). The color variations of the corresponding samples shown in the inset (Figure 4) intuitively reflect the change of the spectra mentioned above. These results could be explained in terms of the influence of salt and temperature on colloidal stability. With an increase of salt concentration, the zeta potentials become lower due to the decrease in the thickness of the double layer, leading to enhancing the aggregation for both AuNPs-PNIPAM and AuNPs. In addition, this aggregation is further enhanced with increasing temperature due to the reduction of the adsorbed charge species and the increase probabilities of collision [33,34]. Consequently, the shoulder peak ranging from 600 to 800 nm is gradually apparent with increasing temperature for both AuNPs-PNIPAM and AuNPs, and this change is enhanced by the increase of NaCl concentration. Differently, this change is suppressed in AuNPs-PNIPAM compared to AuNPs. We believe this inhibitory action is possibly related to the barrier formed by the strong interaction among immobilized PNIPAM molecules. This barrier prevents not only the desorption of citrate anion from the surface of AuNPs caused by increasing temperature, but also the diffusion of salt toward the surface of AuNPs. Notably, the SPR spectrum of AuNPs-PNIPAM solution containing higher concentration of salt even at 36 °C is almost the same as that without salt (see Figure 4C). This particular stability may be related to the increasing negative charges of AuNPs-PNIPAM due to the binding of chloride ions to NH groups of the immobilized PNIPAM molecules [35]. In addition, the peak intensity of SPR spectrum for AuNPs- PNIPAM under higher salt concentration is linearly related to temperature (see Appendix A), indicating that AuNPs-PNIPAM aggregation is mainly caused by the electrostatic effect rather than by the change of PNIPAM conformation. Furthermore, the change in UV spectra ranging from 200 to 300 nm related to the change of citrate is similar to that in Figure 2, and the detailed information can be found in Appendix A.

We also investigated the UV-Vis spectrum change with temperature for AuNPs-PNIPAM/PNIPAM containing NaCl (16.9 mM). As shown in Figure 5A, there is a similar variation trend of the spectrum with temperature to that of AuNPs-PNIPAM/PNIPAM (Figure 3A), further indicating that this thermo-sensitive change has to do with free PNIPAM. However, it is noteworthy that the SPR peak for AuNPs-PNIPAM/PNIPAM containing NaCl is more remarkable than that of AuNPs-PNIPAM/PNIPAM (Figure 3A) when the temperature is above its LCST. The absorption throughout the wavelength range enhances for AuNPs-PNIPAM/PNIPAM containing NaCl, the absorption at the shorter wavelength, however, is stronger than that at a longer wavelength for AuNPs-PNIPAM/PNIPAM. As well-known, salt promotes the formation of a hydrophobic conformation of PNIPAM above its LCST [12] Anion Cl^−^ destabilizes the hydrogen bonding between the amide and water through polarization [35] resulting in that the size of the hydrophobic PNIPAM aggregate above the LCST is larger than that in absence of NaCl [36] For AuNPs-PNIPAM/PNIPAM containing NaCl, the overall enhancement of absorption attributes to the wavelength-independent Mie scattering. For AuNPs-PNIPAM/PNIPAM, the stronger absorption at a shorter wavelength and weaker absorption at a longer wavelength ascribes to wavelength-dependent Rayleigh scattering for AuNPs-PNIPAM/PNIPAM above LCST [37] As shown in Figure 5B, in the presence of NaCl, the variation trend of the absorption spectra with temperature for the PNIPAM solution containing NaCl is similar to that for AuNPs-PNIPAM/PNIPAM solution, not only verifying the deduction mentioned above, but also further indicating the thermo-sensitivity of AuNPs-PNIPAM/PNIPAM arising from free PNIPAM.

Based on all the above-discussed results, the possible mechanisms on the dispersive behavior of AuNPs-PNIPAM in the presence and absence of salt with change in temperature are shown in Figure 6.

### 3.3. The Effect of Proton Diffusion on the Stability of AuNPs-PNIPAM

Figure 7A,B presents the UV-Vis spectrum evolution with time after the addition of HCl in Au-PNIPAM and Figure 7C shows the corresponding change in maximum absorption wavelength with time. After the addition of HCl in the system, the featured peak of AuNPs located at 526 nm firstly experiences a red-shift (Figure 7A) within 0~2 h and then a blue-shift (B) within 2~7 h. This variation is directly illustrated in Figure 7C. The color of the sample varies with time from red to purple, gray, and then to purple (inset in Figure 7), which corresponds to the red-shift and blue-shift of the spectrum. The literature has not yet reported this unexpected finding. The above spectrum shifts and the color variation with time indicates that Au-PNIPAM particles appear firstly to gather together and then to re-disperse. It possibly has relevance to the change in the degree of protonation of Au-PNIPAM with time after adding HCl. Although the carboxyl in the other end group of the synthesized PNIPAM-SH could be responsive to the pH, the obvious response of Au-PNIPAM to the pH is probably not mainly from these carboxyl groups due to its very limited relative content. It is likely to be related to the role of PNIPAM itself. In accordance to the report [38,39], the amide O of PNIPAM chains could accept protons to yield protonated PNIPAM chains. Our experimental results also prove this point. Figure 7D gives acid titration curves for pure water and pure PNIPAM solution. The decreasing trend of pH with increasing HCl for PNIPAM aqueous solution is weaker than that for pure water. There is no doubt that PNIPAM solution could buffer pH decrease through the absorbing proton. When the protons diffuse into the PNIPAM palisades, the net charge of the negative charged Au-PNIPAM reduces, triggering the aggregation of Au-PNIPAM due to the reduction of the electrostatic repulsion. Accordingly, the characteristic peak of SPR has a red-shift. With continual protonation, the negatively charged Au-PNIPAM could become the positive charged one at some point, and after that, the net charge of Au-PNIPAM would gradually increase. Consequently, the repulsion among Au-PNIPAM progressively increases, which results in the re-dispersion of aggregated Au-PNIPAM, and the characteristic peak of SPR resultantly shifts toward the blue wavelength. As expected, this conjecture is confirmed by the change of Zeta potential with time. As shown in Appendix A, the Zeta potential values of Au-PNIPAM are −28.1, −4.6, −3.6, and +5.15 eV, respectively, at 0, 2.5, 4, and 7 h after introducing HCl. Differently, after adding HCl for 1 min, the color change for AuNPs becomes blue, and the peak in the range 600–900 nm appears in the corresponding spectrum (Appendix A). The remarkable difference between AuNPs and AuNPs-PNIPAM in the effect of protons on their stability suggests that the interaction among PNIPAM molecules tethered on the AuNPs surface is strong to inhibit the diffusion of protons. The proposed mechanism on the variation of AuNPs-PNIPAM Zeta potential with time in HCl medium is shown in Figure 8.

We collected the absorption spectra of Au-PNIPAM in the presence of 0.018 M HCl upon heating-and-cooling cycles, and the corresponding result shows in Figure 9. The wavelength at maximum absorption has a reversible change with heating-and-cooling. Considering the facts, Au-PNIPAM does not have the thermo-sensitivity and the PNIPAM can absorb protons; this reversible thermo-responsiveness shown in Figure 9 should derive from the reversible proton diffusion with the temperature change. PNIPAM could adsorb and release protons with decreasing and increasing temperature [40], which causes a decrease and an increase in the zeta potential. As for the wavelength at maximum absorption reversibly changing induced by heating and cooling, it is relevant to the effect of the refractive index on the SPR shift. From a lower to a higher temperature, the refractive index at the local environment of AuNPs increases due to the desorption of the protons absorbed into the PNIPAM palisades, bringing about a red-shift and broadening of the AuNPs SPR [41,42] and vice versa. Here, it should not be neglected that chemical entities may also be responsible for the SPR peak red-shift and the strong plasmon through changing the hot electron lifetime [43]. With increasing the cycles of heating and cooling, the absorbance of the spectrum gradually declines may relate to the precipitation of aggregated AuNPs-PNIPAM.

### 3.4. The Effect of Ethyl Alcohol Diffusion on the Stability of AuNPs-PNIPAM

The variation in UV-Vis spectra with time for AuNPs and AuNPs-PNIPAM solution after the addition of ethyl alcohol are shown in Figure 10A,B. For AuNPs solution after the addition of ethyl alcohol, the intensity of the SPR peak (Figure 10A) around 526 nm gradually decreases with time. At the same time, the peak in the range 600–900 nm appears and gradually red-shifts with time. This is because the diffusion of ethanol causes the decrease of dielectric constant in the local environment of AuNPs, resulting in the continuous formation of AuNPs aggregates due to a gradually decreased electrostatic repulsion of negatively charged AuNPs [44].

For the Au-PNIPAM dispersion in the presence of ethanol (Figure 10B), the SPR spectrum scarcely changes within an hour until 4 h. According to the fact that a decreased dielectric constant can reduce electrostatic repulsion, the introduction of ethanol would also cause aggregation of Au-PNIPAM. However, this phenomenon does not appear. It suggests that the interaction among PNIPAM chains are so strong that ethyl alcohol is hard to diffuse into the palisades of PNIPAM falling to decrease the dielectric constant in the local environment of AuNPs. This strong interaction arises from the hydrogen bonding among adjacent PNIPAM chains or those with water forming an intensive network [21,25]. Although the replacement of water molecules by ethanol is not easy, water molecules in the PNIPAM palisades may be partially displaced by ethanol molecules due to the concentration difference. After the addition of ethanol 4 h, PNIPAM chains locate in an ethanol–water mixed solvent, resulting in a decrease in the SPR intensity due to the reduction of the refractive index. However, the peak in the range 600–900 nm does not appear after 4 h, implying that the aggregation of AuNPs-PNIPAM does not occur. We speculate that the diffusion of water and ethanol molecules in the palisades of PNIPAM are limited. If ethanol molecules completely fully fill in the palisades of PNIPAM, the Zeta potential of AuNPs-PNIPAM and the solubility of PNIPAM [45,46], will evidently decrease, inevitably causing the typical absorbance peak at 600–900 nm designated to the aggregation of AuNPs-PNIPAM. This situation does not meet our experimental results. The most reasonable explanation for this is that there is a rich network of hydrogen bonds existing in PNIPAM palisades on the surface of Au-PNIPAM, preventing both water and ethanol molecules from diffusion. This is also the key reason why AuNPs-PNIPAM does not have typical PNIPAM temperature sensitivity.

### 3.5. The Effect of CTAB Diffusion on the Stability of AuNPs-PNIPAM

The effect of CTAB diffusion on the stability of AuNPs-PNIPAM was investigated, and the corresponding UV-Vis spectra are shown in Figure 11. The UV-Vis spectrum for the AuNPs solution with added CTAB presents a shoulder peak ranging from 600 to 900 nm beside the SPR peak at 526 nm (Figure 11A) within 30 min. In this case, the positively charged heads of CTAB molecules adsorb onto the negatively charged AuNPs, causing the AuNPs aggregation due to reducing electrostatic repulsion among AuNPs [47]. Whereas the UV-Vis spectrum scarcely changes for Au-PNIPAM (Figure 11B). Apparently, it is associated with the fact that the compact palisades of PNIPAM restrain the interaction of the CTAB positive heads with negative AuNPs. In addition, it indicates that the positive heads of CTAB also do not adsorb onto the hydrophilic moieties of the outer edge of PNIPAM palisades. Appendix A shows the UV-Vis spectrum varying with temperature for the AuNPs-PNIPAM solution adding CTAB. The spectrum does not respond to temperature change, revealing the failure of the conformation change of anchored PNIPAM in this situation. With the temperature above the LCST, if the change in the conformation of the immobilized PNIPAM molecules occurs, the alkyl chains of CTAB would interact with the hydrophobic segments located at the outer edge of PNIPAM palisades. Inevitably, the adsorbed positively charged CTAB molecules would reduce the net charge of AuNPs-PNIPAM, thus triggering the aggregation of AuNPs-PNIPAM. The experimental facts, which are entirely different from the above conjecture, once again show that the immobilized PNIPAM on AuNPs does not have the function of thermo-sensitive conformational transformation similar to that of free PNIPAM.

## 4. Conclusions

On account of contradictions regarding the thermo-responsiveness of Au-PNIPAM in the reported literature, the thermo behavior of AuNPs anchored with PNIPAM has been systematically studied. In summary, the following conclusions can be drawn. (1) The temperature sensitivity of PNIPAM immobilized on the AuNPs surface is difficult to function, which makes the AuNPs SPR change driven by the temperature sensitivity of the immobilized PNIPAM very limited. On the contrary, the immobilized PNIPAM shell has an effect of restraining aggregation of AuNPs-PNIPAM. (2) The AuNPs-PNIPAM system containing free PNIPAM shows a clear thermo-responsiveness, which ascribes to the conformation change of free PNIPAM rather than that of the PNIPAM tethered on the surface of AuNPs. Moreover, this effect of free PNIPAM in the AuNPs-PNIPAM solution is only manifested in the turbidity change, and has little effect on SPR of the AuNPs. (3) The failed thermo-responsiveness of AuNPs-PNIPAM composites is due to the strong interaction between PNIPAM molecules anchored on the surface of AuNPs to hinder the conformation change of PNIPAM. (4) The reversible adsorption and desorption of protons by PNIAPM could make the color variation of Au-PNIPAM composite reversible. These findings are very valuable not only to clarify the inconsistent results reported in the literature but also to design functionalized AuNPs-PNIPAM sensing systems for various applications.

## Figures and Tables

**Figure 1 materials-14-00443-f001:**
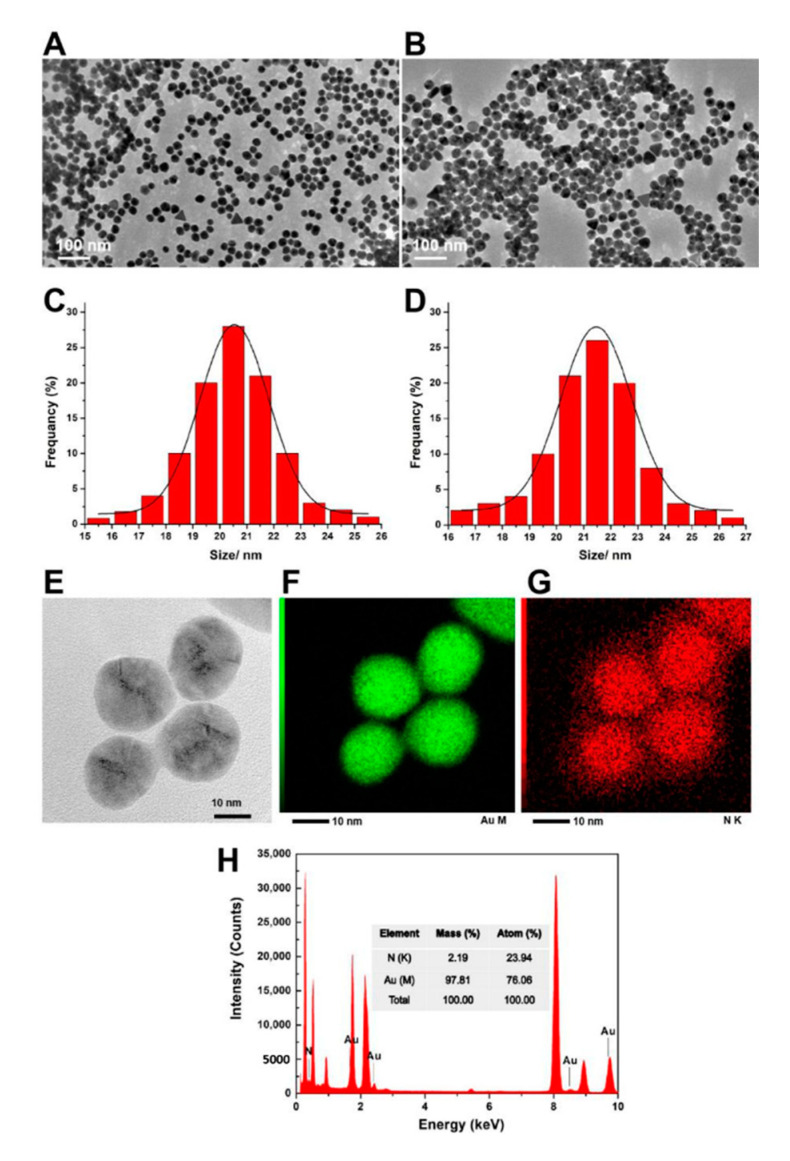
The TEM images for gold nanoparticles (AuNPs) (**A**) and AuNPs-poly (N-isopropylacrylamide) (PNIPAM) (**B**), and size distribution for AuNPs (**C**) and AuNPs-PNIPAM (**D**). High magnification TEM image for Au-PNIPAM (**E**). TEM EDS mapping images for the Au element (**F**) and N element (**G**) and corresponding spectrum (**H**).

**Figure 2 materials-14-00443-f002:**
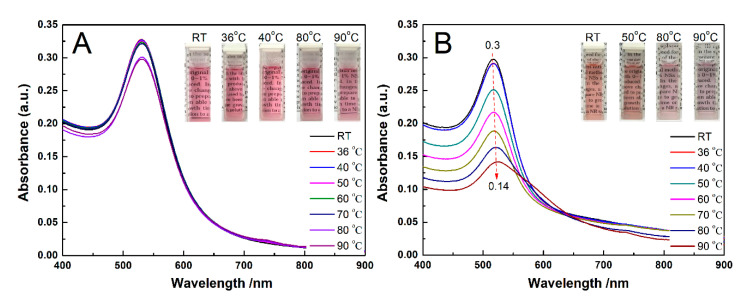
The change in the UV-Vis spectra with temperature for AuNPs-PNIPAM (**A**) and AuNPs (**B**).

**Figure 3 materials-14-00443-f003:**
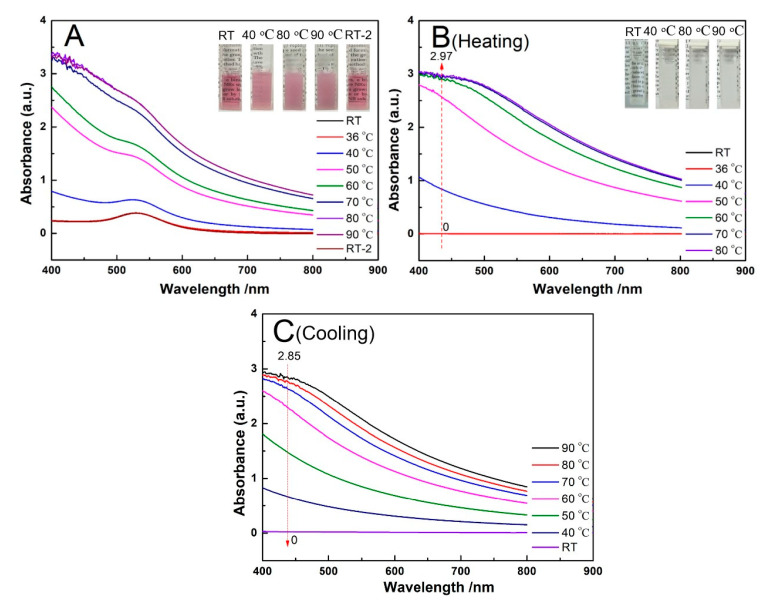
The change in the UV-Vis spectra with temperature for AuNPs-PNIPAM/PNIAM (**A**) and the supernatant from centrifugation of AuNPs-PNIPAM/PNIAM (**B**,**C**).

**Figure 4 materials-14-00443-f004:**
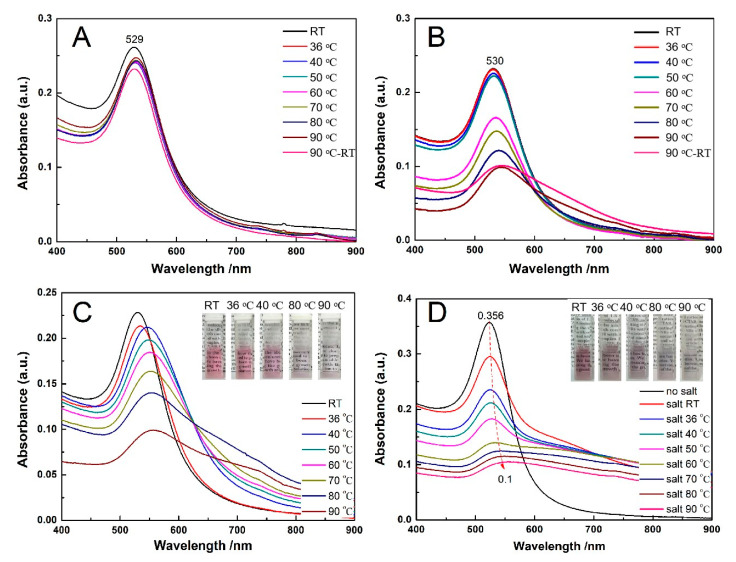
The change in the UV-Vis spectra with temperature under a lower salt concentration (3.2 mM) for AuNPs-PNIPAM (**A**) and AuNPs (**B**), and under a higher salt concentration (16.9 mM) for AuNPs-PNIPAM (**C**) and AuNPs (**D**).

**Figure 5 materials-14-00443-f005:**
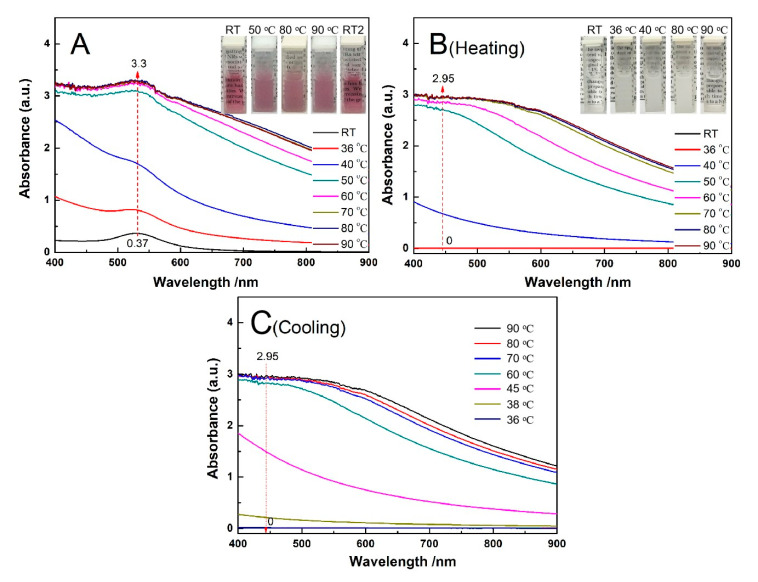
The change in the UV-Vis spectra with temperature for AuNPs-PNIPAM/PNIAM (**A**) and the supernatant from centrifugation of AuNPs-PNIPAM/PNIAM (**B**,**C**) in the presence of 16.9 mM of NaCl.

**Figure 6 materials-14-00443-f006:**
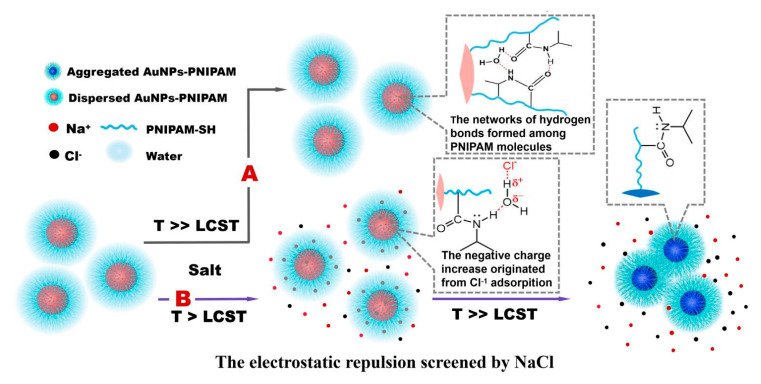
The possible mechanisms on the dispersive behavior of AuNPs-PNIPAM with change in temperature. (**A**) AuNPs-PNIPAM in absence of salt medium. The networks of hydrogen bonds formed among PNIPAM molecules restraining the thermo-sensitivity conformational change of PNIPAM; (**B**) AuNPs-PNIPAM in the presence of salt medium. The dispersive behavior of AuNPs-PNIPAM affected by the temperature dependent electrostatic interactions (including NaCl electrostatic screening, the negative charge increase due to Cl^−1^ adsorption and enhancing dehydration of PNIPAM chains due to Cl^−1^ polarization).

**Figure 7 materials-14-00443-f007:**
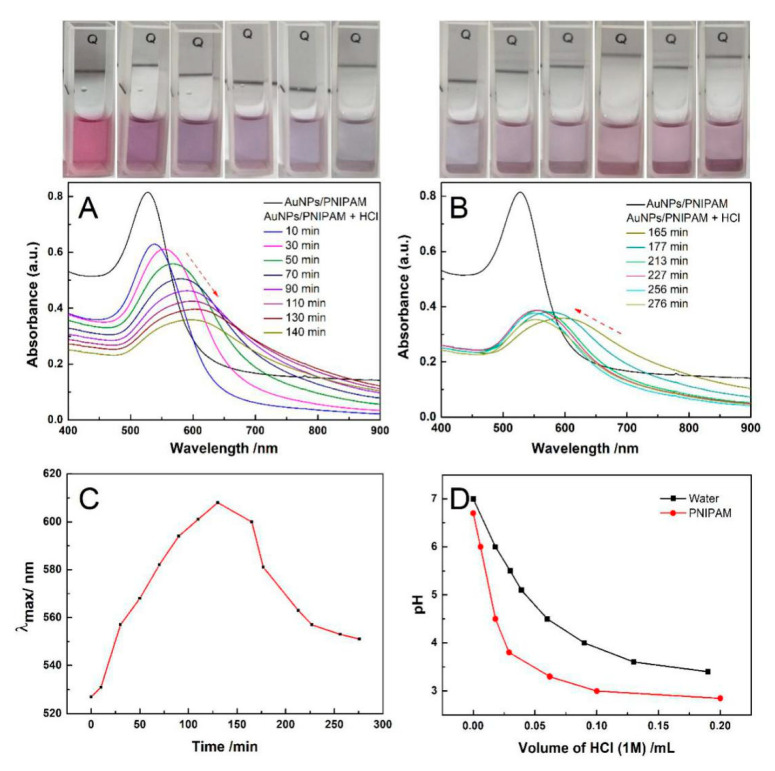
The change in the UV-Vis spectra with time for AuNPs-PNIPAM after addition of HCl (0.002 M) (**A**,**B**), the corresponding change in λ_max_ with time (**C**), and acid titration curves for pure water and PNIPAM solution (3.47 × 10^−6^ mg/L mM) (**D**).

**Figure 8 materials-14-00443-f008:**
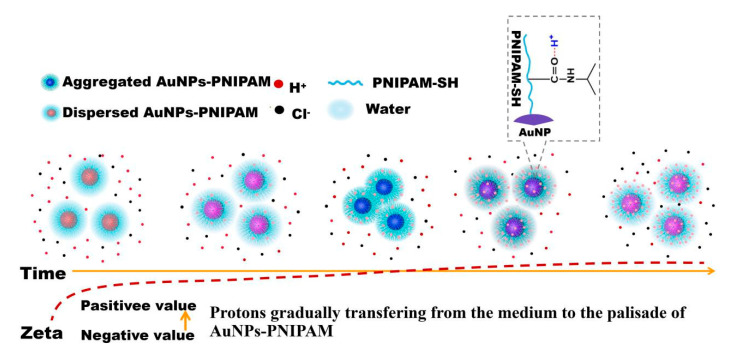
A possible mechanism on the variation of AuNPs-PNIPAM Zeta potential with time in HCl medium. In this process, protons slowly enter into PNIPAM palisades due to protons binding on amide O of PNIPAM, which results in the aggregation of AuNPs-PNIPAM caused by a decrease in Zeta potential. Further increase of protons in PNIPAM palisades makes negatively charged AuNPs-PNIPAM positive, forming the hydrated PNIPAM shell to inhibit the aggregation.

**Figure 9 materials-14-00443-f009:**
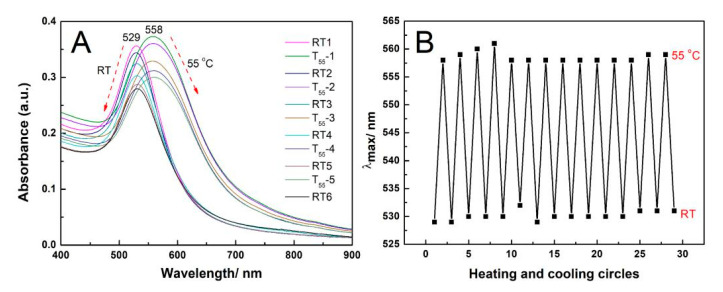
The change in Uv-vis spectra of Au-PNIPAM in the presence of 0.018 M HCl with heating-and-cooling cycles (**A**), and change of λ_max_ with heating-and-cooling cycles (**B**).

**Figure 10 materials-14-00443-f010:**
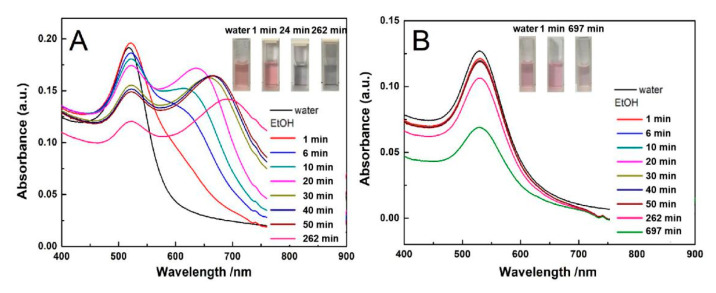
The change in Uv-vis spectra with time for AuNPs (**A**) and Au-PNIPAM (**B**) in ethanol medium.

**Figure 11 materials-14-00443-f011:**
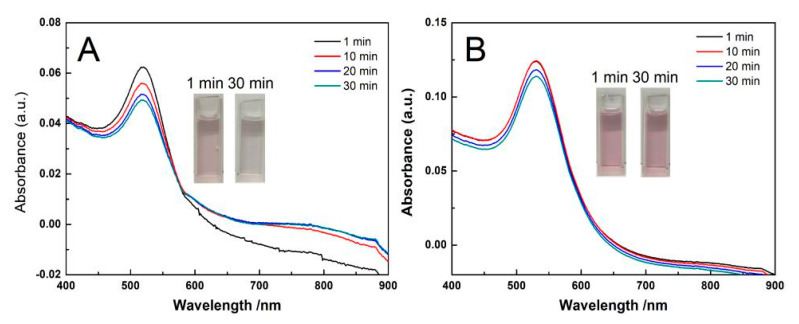
The change in UV-Vis spectra with time for AuNPs (**A**) and Au-PNIPAM (**B**) in 0.025 mM of CTAB solution.

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
