# Peer review of "Conformational Stability of Poly (N-Isopropylacrylamide) Anchored on the Surface of Gold Nanoparticles"

_materials, 2021, doi:10.3390/ma14020443_

Round 1

Reviewer 1 Report

The manuscript is interesting and is related to the temperature sensitivity of hybrid AuNPs-PNIPAM systems.

As the authors wrote in the conclusion, the reported results could really contribute to the: A) clarification of the inconsistent results reported in the literature; B) design of functionalized AuNPs-PNIPAM sensing systems for various applications. 

Nevertheless, the inconsistent results reported in literature are probably the consequence of an insufficient morphological and structural analyses of these intriguing but complex nano systems. This is also the main weakness of this manuscript. The authors reported some morphological data in the supplemental files, but their quality and related discussion are very poor. The size distribution is not convincing for various reasons, the first of which is that the sum of the reported frequencies seems to be something different from 100%

Nothing is report about the 'structure' of these systems that could significantly affect the conformational stability of PNIPAM anchored on the AuNP surface.

The optical characterisation, used alone without any really accurate correlation with morphological and/or structural data, cannot be considered adequate to give a sufficient experimental support to the authors' claims about the behaviour of the sensitivity to temperature of the investigated materials.

In such a context, the authors are suggested to:

1) Expand and improve the TEM data, moving, in any case, them from the supplemental file to the manuscript;

2) Add some structural analysis (X-Ray analysis and/or electron diffraction; Raman; etc.) able to give additional information about the crystallinity and the presence of amorphous(-like) materials that could significantly influence the reported behaviour of the investigated samples in terms of temperature sensitivity.

In such a context, my recommendation is for a major revision.

Author Response

Referee: 1

Comments:

  1. As the authors wrote in the conclusion, the reported results could really contribute to the: A) clarification of the inconsistent results reported in the literature; B) design of functionalized AuNPs-PNIPAM sensing systems for various applications. 

Nevertheless, the inconsistent results reported in the literature are probably the consequence of insufficient morphological and structural analyses of these intriguing but complex nano systems. This is also the main weakness of this manuscript. The authors reported some morphological data in the supplemental files, but their quality and related discussion are very poor. The size distribution is not convincing for various reasons, the first of which is that the sum of the reported frequencies seems to be something different from 100%

Response:

We agree with your suggestion. The complicity in the nanocomposite system of AuNPs-PNIPAM is the main reason for inconsistencies that existed in the reported literature. Thus, there are many factors impacting the UV-Vis spectrum of AuNPs-PNIPAM, such as the length of polymers, the modification density, the size and morphology of AuNPs. In principle, it is necessary to take these factors into full account in this study. From this perspective, we totally agree with you. The introduction in this study as a general background mainly aimed to state that there are inconsistent results in AuNPs-PNIPAM involved reports and in the explanation for the effect of PNIPAM anchored on the surface of AuNPs on the SPR variation. Objectively speaking, however, it is hard to consider all the above factors in an article. Therefore, we just take a special system, given 30 nm of AuNPs saturatedly loaded with PNIPAM, as a case, to be expected to obtain inspiration. That is, strong intermolecular interaction of dense-loaded PNIPAM molecules inhibiting their conformational change, which was not conducive to develop their thermo-sensitive function. This conclusion suggests researchers who expect to develop AuNPs-PNIPAM-based sensors through the conformational change of PNIPAM should take this inspiration into account. Although the system studied in this paper is only a typical one, it has a universal reference significance for the system of AuNPs intensively modified with PNIPAM. This is the key idea of this paper and is detailedly illustrated in the paper, and we hope not to cause misunderstanding to readers.  

We agree with your suggestion for poor quality and discussion of data in the supplemental files. In the revised manuscript, necessary illustrations for data in supporting materials were added and marked. The result of the size distribution was improved. It should be noted that the statistic size was obtained from TEM images. To verify the core-shell structure of AuNPs-PNIPAM, TEM mapping was conducted, and corresponding results were added in the revised manuscript.

  1. Nothing is reported about the 'structure' of these systems that could significantly affect the conformational stability of PNIPAM anchored on the AuNP surface.

The optical characterization, used alone without any really accurate correlation with morphological and/or structural data, cannot be considered adequate to give a sufficient experimental support to the authors' claims about the behaviour of the sensitivity to temperature of the investigated materials.

Response:

Thanks for your suggestions. Since the SPR of AuNPs is sensitive to the local environment, Uv-vis spectroscopy is an effective means to detect the surroundings of AuNPs.

To illustrate the PNIPAM chains densely anchored on the surface of AuNPs are less liable to have a conformational change, the effects of various factors on the SPR for AuNPs and AuNPs-PNIPAM were compared to obtain self-consistent results supporting our viewpoint.   

  • Temperature effect. The conformational change of PNIPAM must be accompanied by a transition from hydrophilic to hydrophobic, which could result in the change of AuNPs-PNIPAM dispersion state and the increase of absorbance of Uv-vis spectrum over 600 nm. For the Au-PNIPAM system which was absent of free PNIPAM, SPR didn't present a distinct change, and in particular, there was no obvious absorption peak above 600 nm, indicating a failed conformational change. For the AuNPs-PNIPAMsystem in presence of free PNIPAM, the Uv-vis spectrum exhibited a strengthened absorbance intensity and no distinct change in the SPR peak position, and actually, the spectrum is just a superposition of the spectrum of non-temperature sensitive AuNPs-PNIPAM and that of temperature-sensitive free PNIPAM. In addition, if the conformation of immobilized PNIPAM and free PNIPAM simultaneously changed when the temperature was higher than LCST, hydrophobic interaction between free and immobilized PNIPAM would inevitably lead to more significant aggregation of AuNPs-PNIPAM. However, no experimental results like the above assumption occurred, still suggesting a failed conformational change of immobilized PNIPAM.
  • Salt effect. At a lower salt concentration, the temperature had no significant effect on the UV-Vis spectrum of AuNPs-PNIPAM. At a higher salt concentration, the results indicated that both effects of salt and temperature on the spectra for AuNPs-PNIPAM with or without free PNIPAM was mainly ascribed to the change in Zeta potential rather than the hydrophobic interaction among immobilized PNIPAM molecules. It suggests the anchored PNIPAM chains are less able to have conformational change.
  • Acid effect. It was found that AuNPs-PNIPAMexhibit reversible thermosensitivity at the acid conditions and this reversibility corresponds to the reversible change in Zeta The results further indicate that the loaded PNIPAM has high conformational stability.
  • Ethanol effect. The experimental results show that in an ethanol medium, severe aggregation occurs in AuNPssolution, while the Uv-vis spectrum of AuNPs-PNIPAM hardly ever changes within 4h. The former was due to the reduced absolute value of zeta potential for AuNPs due to the lower dielectric constant of ethanol, while the latter was related to the stronger interaction between loaded PNIPAM molecules. This strong interaction makes it difficult for ethanol to diffuse in the PNIPAM layer. If ethanol diffusion was easy, the conformation of PNIPAM would change because ethanol is a poor solvent of PNIPAM, and the electrostatic interaction between AuNPs-PNIPAM is thus weakened. These two effects, the lower dielectric constant of ethanol and a poor solvent of PNIPAM, all facilitate the aggregation of AuNPs-PNIPAM. However, the experimental results which are opposite to the assumption showed that there was a strong interaction between the loaded PNIPAM molecules, which is mainly responsible for the stability of tethered PNIPAM.
  • CTAB effect. CTAB is a positively charged surfactant. If AuNPs-PNIPAMrespond to temperature changing from hydrophilic to hydrophobic, hydrophobic PNIPAM would interact with the hydrophobic tail of CTAB when the temperature above LCST. This could lead to a decrease of Zeta potential, causing the aggregation of AuNPs-PNIPAM. However, our experimental results are opposite to the assumption, verifying the stable conformation of loaded PNIPAM.

In this paper, the conformational stability of densely loaded PNIPAM was concluded on the basis of the experimental results of the above five aspects. 

  1. 3. Expand and improve the TEM data, moving, in any case, them from the supplemental file to the manuscript;

Response:

We agree with your suggestion. In the revised manuscript, new TEM data were provided and moved to the manuscript. To verify the core-shell structure of AuNPs-PNIPAM, TEM mapping was conducted. There is a matching spatial distribution for nitrogen element and gold element, indicating the existence of PNIPAM and the core-shell structure of AuNPs-PMIPAM. The results and their corresponding explanations are highlighted in red in the text of the revised manuscript.

  1. 4. Add some structural analysis (X-Ray analysis and/or electron diffraction; Raman; etc.) able to give additional information about the crystallinity and the presence of amorphous(-like) materials that could significantly influence the reported behavior of the investigated samples in terms of temperature sensitivity.

Response:

Thanks for your suggestions. The suggestion you gave us may have something to do with the vagueness of illustration for the principle of PNIPAM thermosensitivity in the manuscript. This study focused on the thermosensitive behavior of AuNPs modified with thermosensitive PNIPAM. PNIPAN (Poly(N-isopropyl acrylamide)) is a typical temperature-sensitive polymer, which is ascribed to its rich hydrophilic amid group and hydrophobic isopropyl. At a lower temperature, water molecules interact with amide forming hydrogen bonds, and PNIPAM adopts a soluble stretched conformation. When the temperature above its LCST (Lower Critical Solution Temperature), water molecules break away from polymers due to a weakened hydrogen bonding and the hydrophobic interaction among isopropyl groups enhances, resulting in a conformation transition from stretched to coiled.

In this study, based on the principle, the sensitivity of Uv-vis spectrum of AuNPs to the local environment, the spectra of AuNPs-PNIPAM at various conditions were systematically investigated, looking forward to acquiring relevant information to infer the conformation change of PNIPAM. All experimental results of the above-mentioned five independent systems are self-consistent, supporting the conclusion that anchored PNIPAM has conformational stability. The correlation between crystallinity or amorphous structure and AuNPs-PNIPAM is a new perspective and we are very thankful for your constructive suggestion. However, due to the limited length of this paper, we hope to further study the effect of crystallinity or amorphous structure on the temperature sensitivity of AuNPs-PNIPAM system in the future.

Reviewer 2 Report

The manuscript is dedicated to the study of behaviour of AuNPs-PNIPAM system in response to the changing conditions in the environment. The study is condacted on a high level and deserves to be published.

I have a couple of comments/questions:

1) On page 6 authors, with a reference to previous studies, suggest that citrate ions can decompose in water at temperatures higher than 30C. This I don't fully understand as the synthesis is conducted in 100C to 90C water solution. Shouldn't then all citrate ions be decomposed already during synthesis and maybe even prior to final Au nps formation?

2) Describing Fig. 9A authors claim that shift during heating is due to the change of the refractive index of the local enviornment of nps and this change is due to the protonation of PNIPAM molecules. The shift of almost 30 nm corresponds to a rather serious change of the refractive index. Are authors sure that this is possible just due to the protonation? Especially if compared to the effect of ethanol (or better say it's absence). One can expect that swithching from water to water-ethanol mixture should change refractive index much stronger, than just protonation. Maybe protonation with subsequent reversable aggregation is more plausible explanation of the effect shown on figure 9A?

Author Response

Referee: 2

Comments:

  1. On page 6 authors, with a reference to previous studies, suggest that citrate ions can decompose in water at temperatures higher than 30C. This I don't fully understand as the synthesis is conducted in 100C to 90C water solution. Shouldn't then all citrate ions be decomposed already during synthesis and maybe even prior to final Au nps formation?

Response: 

Thanks for your reminds. We have revised the corresponding explanation. In this study, AuNPs were prepared following typical Turkevich method. A good dispersed AuNPs solution was obtained by citrate reducing HAuCl4 at 90 oC. In this process, citrate were oxidized. But added citrate were excess and thus could form citrate-capped AuNPs, and there still are abundant free citrate. So, the charged AuNPs has a particularly large Zeta potential in absolute value, which makes the AuNPs extremely stable and has good dispersion even at high temperature. To be noted, for effective displacement of citrate on the surface of AuNPs by SH-PNIPAM, in this study, the prepared AuNPs were centrifuged to remove free citrate and then it was redispersed. This operation causes a decrease in concentration of citrate in the medium, leading to releasing adsorbed citrate moieties towards the medium due to an equilibrium shifting. As a result, the amount of citrate adsorbed on the surface of AuNPs reduces, resulting in unstable AuNPs due to decreased charge density. At the same time, heating could enhance the desorption of citrate on the surface of AuNPs, further strengthening the AuNPs aggregation. For AuNPs-PNIPAM, although the displacement of negatively charged citrate by uncharged PNIPAM makes the Zeta potential of AuNPs-PNIPAM decreased, both the hydration and steric hindrance derived from densely loaded PNIPAM provide AuNPs-PNIPAM a strong stability. Besides, strong interaction among tethered PNIPAM chains inhibit the conformational change responding to temperature. All these factors make AuNPs-PNIPAM more stable than AuNPs in the process of heating.

According to above explanations, we revised the manuscript, and the corresponding modification part was marked in red.

  1. 2.Describing Fig. 9A authors claim that shift during heating is due to the change of the refractive index of the local enviornment of nps and this change is due to the protonation of PNIPAM molecules. The shift of almost 30 nm corresponds to a rather serious change of the refractive index. Are authors sure that this is possible just due to the protonation? Especially if compared to the effect of ethanol (or better say it's absence). One can expect that switching from water to water-ethanol mixture should change refractive index much stronger, than just protonation. Maybe protonation with subsequent reversible aggregation is more plausible explanation of the effect shown on figure 9A?

Response:

Thanks for your remind and we revised the corresponding explanation in newly uploaded manuscript.

Figure 9 shows that at acid condition, SPR peak of AuNPs-PNIPAM red-shifts upon heating and blue-shifts upon cooling. And this shift is reversible. In the study, we provided the literature and experimental results supporting the protonation of PNIPAM chains. Our experimental results shows the decrease in the absolute value of Zeta potential of AuNPs-PNIPAM at acid condition.

In principle, heating will facilitate desorption of protons adsorbed on the PNIPAM, resulting in an increase in absolute value of Zeta potential of AuNPs-PNIPAM. Correspondingly, the SPR peak will blue-shift, but the experimental result is opposite. Similarly, in assumption, cooling will promote protonation of PNIPAM, which will result in the red-shift of SPR peak of AuNPs-PNIPAM, but the experimental result is opposite. Therefore, it is impossible to explain the spectral behavior shown in Figure 9 only in terms of the Zeta potential change.

In fact, the results from Figure 7 and Figure S4 indicates that although the Zeta potential of fully protonated AuNPs-PNIPAM is about +5mV which is much lower than 30 mV (the point at which colloids are tend to aggregate), no shoulder peak appears over 600 nm in the Uv-vis spectrum of AuNPs-PNIPAM in presence of acid. From this perspective, electrostatic interaction is not able to explain the Uv-vis spectrum change for protonated and deprotonated AuNPs-PNIPAM. Therefore, the change of local refractive index at the surface of AuNPs caused by protonated and deprotonated AuNPs-PNIPAM was proposed to explain this spectral behavior. The deprotonation of PNIPAM enhance the interaction among PNIPAM chains on the surface of AuNPs, increasing the refractive index of PNIPAM palisades, which leads to corresponding the red-shift of SPR and an increased absorbance. The protonation of PNIPAM make the distance among anchored PNIPAM chains and solvation increased, decreasing the refractive index of PNIPAM palisades, which leads to corresponding the blue-shift of SPR and a decreased absorbance. The change in refractive index accords with the explanation for the spectral behaviors of Figure 9 and Figure 7.

As you said, the effect of the refractive index on the red-shift of SPR is limited to some extent. Actually, the ligands adsorbed on the surface of AuNPs can change not only the local refractive index but also the electron lifetime of plasmon. There is a study found when suitable chemical entities that can participate in electron transfer with AuNPs are adsorbed on AuNPs surface, it is expected that change in electron lifetime is the phenomenon which is responsible for red-shift and a strong plasmon.[48] The red-shift of the SPR of AuNPs-PNIPAM caused by the transformation between protonation and deprotonation of PNIPAM may be related to both effects of refractive index and electron lifetime. According to the reported literature, combined with our research results, the relevant explanation was revised.

Reviewer 3 Report

The authors discuss why previous studies reported inconsistent results on the temperature-responsive behaviors of AuNPs-PNIPAM. It was successfully found that the above-mentioned inconsistencies derived from the presence/absence of Free PNIPAM, which plays a crucial roll in the AuNPs aggregation instead of surface-bound PNIPAM. Potentially, these results are considered to be highly valuable to the research field of functional nanomaterials.

On the other hand, the logical structure of the paper is so redundant that readers will find difficulties to extract messages from the manuscript. I think the paper should not be accepted without substantial reconstructions of the logical composition.

In particular, the following points need to be modified:

1. Clarify the approach of this research. What is new in this research compared with previous reports which suggest the importance of free PNIPAMs.
2. Explain what each figure "means" in its caption. What can be concluded from each figure? Only "what was done" are shown in the current manuscript.
3. Too many figures are given for the amount of the contents. I think some of the figures should be shown together or moved to the supporting information.
4. In the section of "Conclusion", please explain from which experiments each conclusion is obtained.
5. There are several colloquial or redundant expressions. It is recommended to use an academic English proofreading service.

Author Response

Referee: 3

Comments:

  1. Clarify the approach of this research. What is new in this research compared with previous reports which suggest the importance of free PNIPAMs.

Response:

Thanks for your suggestions. Inspired by some inconsistent reports, the influence of related factors on the SPR behavior of a given AuNPs modified with PNIPAM system was investigated. Since the SPR of AuNPs is sensitive to the local environment, Uv-vis spectroscopy is an effective means to detect the surroundings of AuNPs. To illustrate the PNIPAM chains densely anchored on the surface of AuNPs are less liable to have a conformational change, the effects of various factors on the SPR for AuNPs and AuNPs-PNIPAM were compared to obtain self-consistent results supporting our viewpoint.

  • Temperature effect. The conformational change of PNIPAM must be accompanied by transition from hydrophilic to hydrophobic, which could result in the change of AuNPs-PNIPAM dispersion state and the increase of absorbance of Uv-vis spectrum over 600 nm. For the Au-PNIPAM system which was absence of free PNIPAM, SPR didn't present an distinct change, and in particular there was no obvious absorption peak above 600 nm, indicating a failed conformational change. For the AuNPs-PNIPAM system in presence of free PNIPAM, the Uv-vis spectrum exhibited a strengthened absorbance intensity and no distinct change in the SPR peak position, and actually, the spectrum is just superposition of the spectrum of non-temperature sensitive AuNPs-PNIPAM and that of temperature sensitive free PNIPAM. In addition, if the conformation of immobilized PNIPAM and free PNIPAM simultaneously changed when the temperature was higher than LCST, hydrophobic interaction between free and immobilized PNIPAM would inevitably lead to more significant aggregation of AuNPs-PNIPAM. However, no experimental results like above assumption occurred, still suggesting a failed conformational change of immobilized PNIPAM.
  • Salt effect. At a lower salt concentration, temperature had no significant effect on the UV-Vis spectrum of AuNPs-PNIPAM. At a higher salt concentration, the results indicated that both effect of salt and temperature on the spectra for AuNPs-PNIPAM with or without free PNIPAM was mainly ascribed to the change in Zeta potential rather than the hydrophobic interaction among immobilized PNIPAM. It suggests the anchored PNIPAM chains are less able to have conformational change.
  • Acid effect. It was found that AuNPs-PNIPAM exhibit reversible thermosensitivity at acid condition and this reversibility corresponds to the reversible change in Zeta potential. The results further indicate that the loaded PNIPAM has high conformational stability.

(4) Ethanol effect. The experimental results show that in ethanol medium, severe aggregation occurs in AuNPs solution, while the Uv-vis spectrum of AuNPs-PNIPAM hardly ever change within 4h. The former was due to the reduced absolute value of zeta potential for AuNPs due to the lower dielectric constant of ethanol, while the latter was related to the stronger interaction between loaded PNIPAM molecules. This strong interaction makes it difficult for ethanol to diffuse in the PNIPAM layer. If ethanol diffusion was easy, the conformation of PNIPAM would change because ethanol is a poor solvent of PNIPAM, and the electrostatic interaction between AuNPs-PNIPAM is thus weaken. These two effects, lower dielectric constant of ethanol and a poor solvent of PNIPAM, all facilitate the aggregation of AuNPs-PNIPAM. However, the experimental results which are opposite to the assumption showed that there was a strong interaction between the loaded PNIPAM molecules, which is mainly responsible for the stability of tethered PNIPAM.

(5) CTAB effect. CTAB is a positively charged surfactant. If AuNPs-PNIPAM respond to temperature changing from hydrophilic to hydrophobic, hydrophobic PNIPAM would interact with hydrophobic tail of CTAB when the temperature above LCST. This could lead to the decrease of Zeta potential, causing the aggregation of AuNPs-PNIPAM. However, our experimental results are opposite to the assumption, verifying the stable conformation of loaded PNIPAM.

Compared with previous reports, we found that PNIPAM molecules densely loaded on the surface of AuNPs does not have temperature-sensitive conformational change. When free PNIPAM exist in the AuNPs-PNIPAM system, the temperature-sensitive conformational change originate from free PNIPAM rather than immobilized ones. That is, in the system where immobilized PNIPAM and free PNIPAM exist together, the functions of both are independent and do not have obvious interaction.

  1. Explain what each figure "means" in its caption. What can be concluded from each figure? Only "what was done" are shown in the current manuscript.

Response: 

Thanks for your reminds. Following the general presentation way of graph, we gave the captions below the corresponding figures, which were only used to state what figures they were. And relevant experimental conditions were also given when necessary. Considering the limited length of the paper and avoidance of repeated expression, the features of figures presented, the explanations for the phenomena and the conclusion drawn were systematically illustrated in the text, in which readers could acquire comprehensive understand.

  1. Too many figures are given for the amount of the contents. I think some of the figures should be shown together or moved to the supporting information.

Response:

Thanks for your suggestion. In this study, relatively rich experimental data presented are aimed to support the reliability of the conclusion. Considering the closed relevance of the experimental data to the conclusion given, and rationality of the graphs located in the text, we integrated the data, and the total number of graphs in the text was 11, which did not exceed the requirements of the journal.

  1. In the section of "Conclusion", please explain from which experiments each conclusion is obtained.

Response:

Thanks for your rational suggestion. It is note to mention, the study focused on the special conformational stability of PNIPAM densely loaded on the surface of AuNPs. The manuscript illustrated the influences of the following factors including temperature, salt concentration, free PNIPAM, ethanol, acid and CTAB on the SPR behavior of AuNPs-PNIPAM. Because all these results helping to draw the conclusions are correlated, in order to avoid repetition, they were not concluded independently corresponding to every part, instead, giving the main points for the whole paper.

  1. 5. There are several colloquial or redundant expressions. It is recommended to use an academic English proofreading service.

Response:

Thanks for your advice. We read the manuscript carefully and revised the expression as much as we could hoping to meet the reader's reading requirements.

Reviewer 4 Report

In this manuscript entitled “Conformational stability of poly (N3 isopropylacrylamide) anchored on the surface of gold nanoparticles” author present the spectroscopic data for verification the temperature sensitive failure of poly (N-isopropylacrylamide) (PNIPAM) anchored on the surface of gold nanoparticles (AuNPs.  In my opinion, manuscript require the additional information and explanation before publishing.

Suggestions the authors might want to consider:

  1. Firstly, in introduction should be discussed the motivation for presentation of described system of two different form chemical point of view nanoparticles system.
  2. Please provide and discussion data regarding size particles distribution from TEM.
  3. In case a data present and disused in first part of R&D, please also include AuNPs plasmon effects on changes in temperature function. Base on that, verify the present mechanism on fig 5.
  4. Plase dissice and provide data of zeta potential measurements, a specially their mono or polidispersion? On fig 7 present isoelectric point of studied system.
  5. Include and discuss the formation of dielectric layer and their changes in present mechanism on the variation of AuNPs-PNIPAM Zeta potential with time in HCl medium

Based on the above review, the manuscript should be revised. If all shortcomings and drawbacks of the current manuscript are improved, a revised manuscript can be considered for publishing.

Author Response

Referee: 4

  1. Firstly, in introduction should be discussed the motivation for presentation of described system of two different form chemical point of view nanoparticles system.

Response:

Thanks for your suggestion. The plasma of AuNPs which is sensitive to local environmental has been extensively studied. Based on the typical thermosensitive PNIAPM, PNIPAM chains immobilized on the surface of AuNPs (AuNPs-PNIPAM) are also naturally expected to show a similar thermo-responsiveness and thus develop thermo-responsive sensors. Many studies have been reported on this. From a large number of reports, we find that there are many inconsistent reports. In the introduction section, the manuscript listed a large number of such comparative literature. The aim is to show that there are inconsistencies even in the same system in different reports. On the basis of a large number of literature analysis, we believe that the main disagreement in the literature report is whether the PNIPAM loaded on the AuNPs surface still has the same temperature sensitivity as the free PNIPAM. Although there are many factors affecting this conclusion, such as AuNPs particle size, PNIPAN loading density, PNIPAM molecular weight, etc., in general, in any case, PNIPAM loaded on the AuNPs are not easy to undergo conformational changes compared to free PNIPAM. Specially, hydrophobicity variation caused by conformational changes of PMIPAM is a result of the collective behavior of many molecules. Insufficient load is difficult to have a significant thermal-sensitive effect on the dispersion behavior of the AuNPs-PNIPAM. Densely loading diminish the room for conformational change due to strong interaction among adjacent chains. In this study, based on the principle, the sensitivity of Uv-vis spectrum of AuNPs to local environment, the spectra of AuNPs-PNIPAM at various conditions (including temperature, salt concentration, free PNIPAM, ethanol, acid and CTAB) were systematically investigated, looking forward to acquire relevant information to infer the conformation change of PNIPAM. All experimental results of the above-mentioned five independent systems are self-consistent, supporting the conclusion that anchored PNIPAM has conformational stability. It means that dense-loaded PNIPAM does not exhibit thermo-sensitive like free PNIPAM. If we could draw such inspiration from this study, we think we have achieved our goal. The introduction is described following this logic.

  1. Please provide and discussion data regarding size particles distribution from TEM.

Response:

Thanks for your suggestion. In revised manuscript, new TEM data and size distribution were provided and discussed, which were moved to the manuscript.

3.In case a data present and disused in first part of R&D, please also include AuNPs plasmon effects on changes in temperature function. Base on that, verify the present mechanism on fig 5.

Response:

We agree with your suggestion. The mechanism given in Figure 6 was proposed on the basis of the first and second parts of R&D. In original manuscript, it is inappropriate to put it in the second part of R&D. Therefore, at the end of the second parts of R&D, the mechanism proposed in figure 6 is specially described, and it is emphasized that the mechanism is proposed on the basis of the first and second parts of R&D. We hope revised expression could eliminate the misunderstanding.

  1. Plase dissice and provide data of zeta potential measurements, a specially their mono or polidispersion? On fig 7 present isoelectric point of studied system.

 Response: 

Thanks for your suggestion. Because of a large amount of data involved in this study, although we used many combined diagrams to reduce space of layout they occupied, some reviewers still think there are too many diagrams and hope further integrate. As far as the Zeta potential of this study is concerned, we mainly want to use it to support the speculation of the causes of the related phenomena. Objectively speaking, the particle size analysis based on light scattering has many factors affecting the experimental results. Because it is inevitable to input some parameters, such as refractive index of particles, water absorption, etc. For composites, these parameters are completely estimated by person, which seriously affects the reliability of its determination. Our TEM determination indicates that the obtained AuNPs and AuNPs-PNIPAM are uniform in size. Thus, only the Zeta potential values were provide in the manuscript, which did not influence the conclusions drawn of this study.

It is a reasonable suggestion for providing isoelectric point in Figure S4. Nevertheless, for the AuNPs-PNIPAM system with adding HCl, because of the heterogeneous distribution of protons in the system, the pH of the solution may not reflect the real situation of local pH. If the pH determined is given as isoelectric point, it may cause misunderstanding.

  1. Include and discuss the formation of dielectric layer and their changes in present mechanism on the variation of AuNPs-PNIPAM Zeta potential with time in HCl medium

 Response:

We agree with your suggestion. In revised manuscript, we added a simple illustration in the caption of Figure 8 for easy understanding to readers.

Round 2

Reviewer 1 Report

The authors made an appreciated work of revision for addressing my previous criticisms. Convincing further details and explanations are given in their cover letter. 

Nevertheless, a further improvement of the manuscript could be achieved transferring all (or at least the predominant part of) the content of the cover letter into the manuscript, in order to improve its readability. 

Also some parts (the most significant information) of the content of the supplementary files could be moved in the revised manuscript.

Author Response

Referee: 1

Comments:

The authors made an appreciated work of revision for addressing my previous criticisms. Convincing further details and explanations are given in their cover letter. 

Nevertheless, a further improvement of the manuscript could be achieved transferring all (or at least the predominant part of) the content of the cover letter into the manuscript, in order to improve its readability. 

Also some parts (the most significant information) of the content of the supplementary files could be moved in the revised manuscript.

Response:

Thanks for your suggestions. In revised manuscript, the key idea of this paper (corresponding to the response to the first piece of comment in cover letter) was partially transferred into the manuscript at the end of the Introduction part. A brief summary to Figure 2 and Figure 3 (corresponding to the temperature effect, the response to the second piece of comment in cover letter) was also transferred into the manuscript at the end of the the first part of R&D. The discussion to the Figure S2 in supplementary files was partially added into the manuscript at the corresponding locations. All these modifications were marked in red.

Reviewer 3 Report

The logic of manuscript have been improved, especially by addition of Figure 1. The additional description at Introduction and  Figure 8 also contribute readability.

I think the manuscript should be accepted after some minor revisions.

  1. Please provide experimental details about how to obtain elemental mappings (Figure 1F and 1G). EDS? or EELS?
  2. Please show the obtained spectra (EDS or EELS) to make the elemental mappings (Figure 1F and 1G) in the manuscript or supporting information to prove the element (Au and N) is really exist in the meaningful amount. 
  3. In the figure 7, is it possible to eliminate the possibility of partial hydrolysis effect of amide units in the presence of HCl?

Author Response

Referee: 3

Comments:

The logic of manuscript have been improved, especially by addition of Figure 1. The additional description at Introduction and  Figure 8 also contribute readability.

I think the manuscript should be accepted after some minor revisions.

  1. Please provide experimental details about how to obtain elemental mappings (Figure 1F and 1G). EDS? or EELS?

Please show the obtained spectra (EDS or EELS) to make the elemental mappings (Figure 1F and 1G) in the manuscript or supporting information to prove the element (Au and N) is really exist in the meaningful amount. 

Response:

Thanks for your suggestions. TEM EDS spectrum and images of Au-PNIPAM composites were collected with a JEM-2800 in STEM mode. In revised manuscript, these details were added in the Characterization part of Experimental Section and were marked in red. Besides, the TEM EDS spectrum containing the amounts of elements (Au and N) was added in Figure 1.      

  1. In the Figure 7, is it possible to eliminate the possibility of partial hydrolysis effect of amide units in the presence of HCl?

Response:

Thanks for your reasonable suggestion. Indeed, there are studies shown the hydrolysis of PNIPAM could happen under certain conditions, such as in strong alkaline solutions or at higher temperature. KrisztinaLaszlo et al. found that PNIPAM molecules have obvious hydrolysis in a strong alkaline solution [Search for the origin of Discrepancies in Osmotic Measurements of the PNIPAM-Water System, Periodica Polytechnica Chemical Engineering, 2017,61(1), pp.39]. Chia-Hung Chen et al. suggested that the hydrolysis of PNIPAM molecules might occur at temperatures above 180oC [A one-step hydrothermal route to programmable stimuli-responsive hydrogels. Chem. Commun., 2015,51,6617-6620]. Therefore, it could be inferred that the experimental results obtained in this study may not be related to the hydrolysis due to the experimental conditions being neither high temperature nor strong alkaline. Certainly, the notable hydrolysis of PNIPAM may occur if PNIPAM molecules exist in acidic conditions for a long time. This mater as another topic worthy studying, we hope to pay attention to it in the future.
